# Cardiotoxic profiles of CAR-T therapy and bispecific T-cell engagers in hematological cancers
Badri Karthikeyan[1,2], Sunitha Shyam Sunder[2], Igor Puzanov[3], Scott H. Olejniczak [4], Saraswati Pokharel[2] & Umesh C. Sharma [1] ✉

## Abstract

**Background** Chimeric antigen receptor (CAR) T-cell therapy and bispecific T-cell engagers, which redirect T-cells to tumor antigens, have immensely benefitted patients with relapsed/refractory B-cell cancers. How these therapies differ in cardiotoxicity is underexplored. We used the World Health Organization pharmacovigilance database, VigiBase, to compare cardiotoxicity profiles between CD19-targeted CAR-T therapy and blinatumomab (a CD19/CD3-targeted bispecific T-cell engager).

**Methods** Safety reports in VigiBase were filtered for diffuse large B-cell lymphoma (DLBCL, n = 17,479) and acute lymphocytic leukemia (ALL, n = 28,803) for all adverse reactions. Data were further filtered for patients taking CAR-T therapy or blinatumomab. Reporting odds ratios (ROR) and fatality rates were compared between CAR-T cell products (e.g. tisagenlecleucel and axicabtagene ciloleucel), and between CAR-T therapy and blinatumomab.

**Results** Tisagenlecleucel is associated with cardiac failure ($IC_{025}$ = 0.366) with fatality rates of 85.7% and 80.0% in DLBCL and pediatric ALL patients respectively. For DLBCL patients, axicabtagene ciloleucel has greater reporting for hypotension than tisagenlecleucel (ROR: 2.54; 95% CI: 1.28–5.03; p = 0.012), but tisagenlecleucel has higher fatality rates for hypotension than axicabtagene ciloleucel [50.0% (tisagenlecleucel) vs 5.6% (axicabtagene ciloleucel); p < 0.001]. Blinatumomab and tisagenlecleucel have similar fatality rates for hypotension in pediatric ALL patients [34.7% (tisagenlecleucel) vs 20.0% (blinatumomab); p = 0.66].

**Conclusions** Tisagenlecleucel is associated with severe and fatal adverse cardiac events, with higher fatality rates for hypotension compared to axicabtagene ciloleucel in DLBCL patients, but similar hypotension fatality rates compared to blinatumomab in pediatric ALL patients. Effective management necessitates experienced physicians, including cardio-oncologists, skilled in interdisciplinary approaches to manage these toxicities.

## Plain Language Summary

Chimeric antigen receptor (CAR) T-cell therapy and blinatumomab are two new types of cancer therapies used to treat blood cancers that fail to respond to conventional chemotherapy. Our goal is to study if there are major differences in how these treatments affect the heart. We analyzed a large, global database of patients who had these treatments. We find that in a blood cancer called diffuse large B-cell lymphoma, two CAR-T cell therapies are linked to heart failure and low blood pressure. In another type of cancer, acute lymphocytic leukemia, CAR-T cell therapy is associated with heart failure and cardiac arrest. The study suggests that given the frequency and severity of these side effects, clinical care should involve an interdisciplinary team experienced in managing these serious side effects.

Despite improvements in conventional chemotherapy regimens for the management of aggressive B-cell cancers, including acute lymphoblastic leukemia (ALL) and diffuse large B-cell lymphoma (DLBCL), chemotherapy resistance and cancer relapse pose a serious problem, often resulting in poor survival outcomes[1,2]. Chimeric antigen receptor (CAR) T cell therapies have revolutionized the treatment of relapsed or refractory B-cell cancers by inhibiting B-lymphocyte surface antigen B4 (CD19), which is expressed in all B cell lineages[3,4]. Unlike conventional therapies, CAR-T therapy consists of T cells that are engineered to incorporate a synthetic receptor, which can recognize and eliminate cancer cells expressing a target antigen[5,6].

[1]Department of Medicine, Division of Cardiology, Jacobs School of Medicine and Biomedical Sciences, Buffalo, NY 14203, USA. [2]Department of Pathology and Laboratory Medicine, Roswell Park Comprehensive Cancer Center, Buffalo, NY 14203, USA. [3]Department of Medicine, Division of Oncology, Roswell Park Comprehensive Cancer Center, Buffalo, NY 14203, USA. [4]Department of Immunology, Roswell Park Comprehensive Cancer Center, Buffalo, NY 14203, USA. ✉e-mail: sharmau@buffalo.edu

Unfortunately, not all patients are eligible for CAR-T therapy, and this treatment strategy can often result in a poor response due to various factors, such as the overexpression of inhibitory checkpoint molecules (e.g., PD-1) in immune cells[4,7,8]. Immune checkpoint inhibitors (ICIs) that block PD-1 can help prevent CAR-T cell exhaustion and thus improve the effectiveness of CAR-T therapy[9]. However, ICIs are cardiotoxic[10], whereby even short-term treatment of ICIs could result in a proinflammatory cytokine storm and increased production of damage-associated molecular patterns (DAMPs) in myocardial tissues, with increased myocardial expression of MyD88 and NLRP3 promoting fibrosis, hypertrophy, and myocardial inflammation[11]. Bispecific T-cell engagers are antibodies that can simultaneously bind to both T cells (e.g., via CD3) and tumor antigens (e.g., CD19), thereby directly promoting the T-cell killing of tumor cells that is independent of major histocompatibility complex (MHC) expression[12–14]. While these novel treatments offer promising venues for the treatment of hematological malignancies, they may have adverse cardiovascular events that could potentially hinder their use.

The most common and life-threatening toxicity of CAR-T therapy is cytokine release syndrome (CRS), which is characterized by elevated cytokine and inflammatory markers, including interleukin-6 (IL-6)[15]. CRS consists of a collection of adverse events that include fever, hypotension, tachycardia, and hypoxia[16,17]. The severity of CRS is described by a grading scale, with hypotension being a characteristic symptom for CRS of Grades 2 and higher[17–19]. A 2019 registry study found patients with Grade 2 CRS with troponin elevation developed significantly increased risk for future cardiovascular events, including decompensated heart failure and new arrhythmic events[20]. While corticosteroids and tocilizumab, which is an IL-6 receptor antagonist, are used regularly to mitigate CRS symptoms, patients with severe CRS symptoms require intensive care treatment and cessation of CAR-T therapy[21,22]. Unfortunately, there is limited information concerning the onset, severity and overall outcomes of adverse cardiovascular events associated with CAR-T therapy and bispecific T-cell engagers, especially when controlled for the cancers treated. Considering that CAR-T therapy and bispecific T-cell engagers use different drug delivery approaches that redirect T cells to target B-cell surface antigens, a comparative approach using real-world pharmacovigilance databases would offer detailed insights concerning the cardiovascular side-effect profiles of CRS.

In this real-world study, we use a global, retrospective World Health Organization (WHO) database to compare the cardiovascular safety profiles between CAR-T therapy and bispecific T-cell engagers. We find that in patients with diffuse large B-cell lymphoma, tisagenlecleucel is associated with severe adverse cardiac events, including cardiac failure, compared to axicabtagene ciloleucel. In pediatric patients with acute lymphocytic leukemia, we find that tisagenlecleucel is associated with highly fatal cardiac failure and cardiac arrest, both adverse events that are not associated with blinatumomab. Recognition of these life-threatening cardiovascular side effects will help physicians to identify patients at increased risk of complications and facilitate prompt clinical management.

## Methods
### Data source and processing
VigiBase is a global, deduplicated WHO database that contains over 30 million Individual Case Safety Reports (ICSRs) from November 14, 1967 to January 26, 2022 and is maintained by the Uppsala Monitoring Centre (UMC)[23]. Over 130 member countries in the WHO Program for International Drug Monitoring (WHO PIDM) collect and share ICSRs electronically with VigiBase. Each ICSR contains information about patient demographics, indications, adverse reactions, seriousness of side effects, fatality, and time to onset[23]. We uploaded the data from VigiBase into DB Browser for SQLite version 3.12.2 and processed using in-house Structured Query Language (SQL) programming. We then extracted relevant tables as comma-separated values files for further analysis. The Roswell Park Institutional Review Board exempted this study from review given the deidentified nature of VigiBase data. Since the dataset is publicly available with access maintained by the UMC, informed consent was not required.

### Immunotherapies investigated and adverse drug reactions
The medication categories investigated in this study are six CAR-T therapies (tisagenlecleucel, axicabtagene ciloleucel, brexucabtagene autoleucel, lisocabtagene maraleucel, idecabtagene vicleucel, ciltacabtagene autoleucel), and one bispecific T-cell engager (blinatumomab) (Supplementary Table 1). The Medical Dictionary for Regulatory Activities (MedDRA) terminology is the international medical terminology developed under the auspices of the International Council for Harmonisation of Technical Requirements for Pharmaceuticals for Human Use (ICH). MedDRA offers a standardized approach to classify adverse reactions associated with the use of medications using a multiaxial hierarchy model[24]. Therefore, we performed SQL programming to search for ICSRs containing adverse reactions listed under the MedDRA version 24.1 system organ classes of cardiac disorders, vascular disorders, and investigations.

### Information component analyses
Researchers at the UMC utilized a Bayesian confidence propagation neural network in conjunction with principles from information theory to derive the information component (IC), which is a statistic used to detect significant adverse reactions for a given category of medications[25]. The UMC researchers provided an approximation formula that can accurately calculate the information component, as illustrated below (Eq. 1)[26]:

$$IC = \log_2\left(\frac{N_{AD} + 0.5}{N_E + 0.5}\right), \text{ where } N_E = \frac{N_A N_D}{N_{total}} \qquad (1)$$

Where $N_{AD}$ is the observed number of reports in VigiBase for a given drug-adverse reaction pair, $N_E$ is the expected number of reports for a given drug-reaction pair, $N_A$ is the number of reports for a given adverse reaction, $N_D$ is the number of reports for a given drug, and $N_{total}$ is the total number of reports in VigiBase. The $IC_{025}$ represents that lower bound of the 95% credibility interval for IC, where an $IC_{025}$ greater than 0 suggests a statistically significant adverse reaction signal for a given drug[26]. Information components were calculated to evaluate the strengths of adverse reaction signals for CAR-T therapies. However, ICs can only be used to compare a drug-adverse reaction pair against the entire database.

### Statistical analyses
To compare the strengths of reports between two drug categories for a given adverse reaction, reporting odds ratios (RORs) and 95% confidence intervals (CI) were calculated by constructing 2 × 2 contingency tables (Supplementary Table 2), while Pearson's chi-squared test or Fisher's exact test were used to examine statistical significance as appropriate. A p-value of <0.05 was considered statistically significant for all statistical analyses. Statistical analyses were performed using Microsoft Excel (16.47.1, 21032301) and SAS v9.4 (Cary, NC).

### Reporting summary
Further information on research design is available in the Nature Portfolio Reporting Summary linked to this article.

## Results
### Characterization of CAR-T therapy safety reports
Based on the records obtained up to January 26, 2022, a total of 30,039,123 Individual Case Safety Reports (ICSRs) were reported in VigiBase. These ICSRs were filtered into 5,576 reports containing CAR-T therapy for all adverse reactions, and these reports were then subdivided by CAR-T cell product used, which included tisagenlecleucel ($N_D$ = 2,233), axicabtagene ciloleucel ($N_D$ = 3,035), brexucabtagene autoleucel ($N_D$ = 170), lisocabtagene maraleucel ($N_D$ = 82), idecabtagene vicleucel ($N_D$ = 54), and ciltacabtagene autoleucel ($N_D$ = 2), for all adverse reactions. After filtering for cardiac, investigational, and vascular adverse reactions, the number of reports obtained from clinical studies was reduced to n = 667 for tisagenlecleucel, n = 101 for axicabtagene ciloleucel, and n = 7 for brexucabtagene autoleucel, n = 2 for lisocabtagene maraleucel, n = 3 for idecabtagene

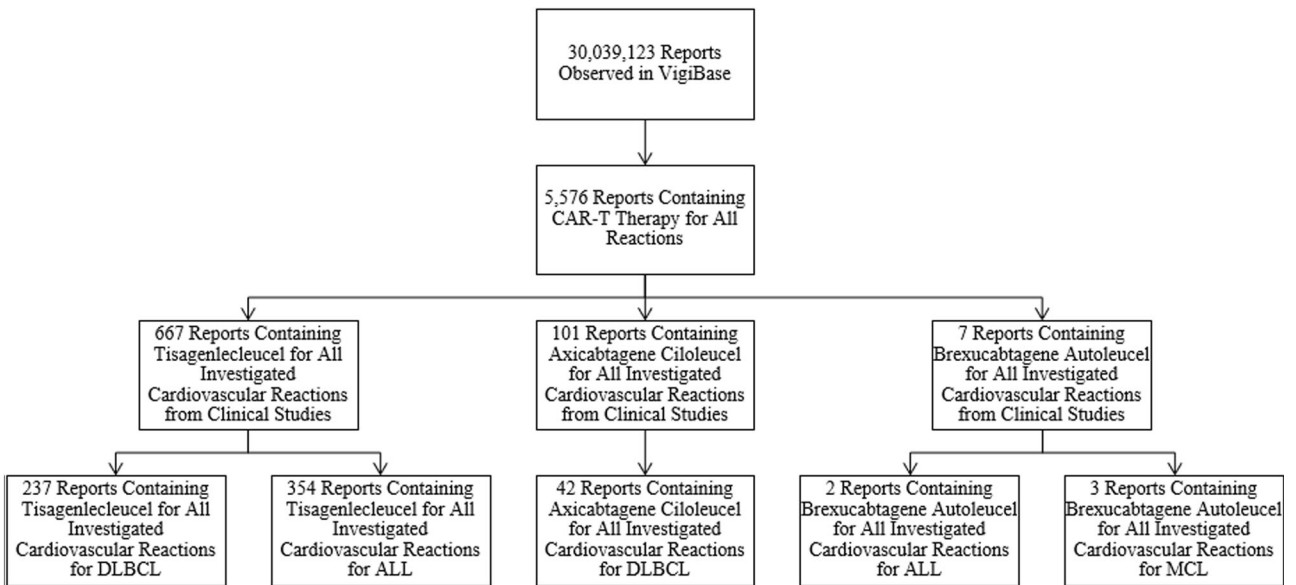

**Fig. 1 | Flowchart illustrating the filtering of Individual Case Safety Reports (ICSRs).** After obtaining all CAR-T therapy ICSRs from VigiBase, the reports were filtered for tisagenlecleucel, axicabtagene ciloleucel, and brexucabtagene autoleucel for all cardiac, vascular, and investigational adverse reactions. These ICSRs were subsequently filtered by cancer indications consisting of diffuse large B-cell lymphoma (DLBCL), acute lymphocytic leukemia (ALL), and mantle cell lymphoma (MCL).

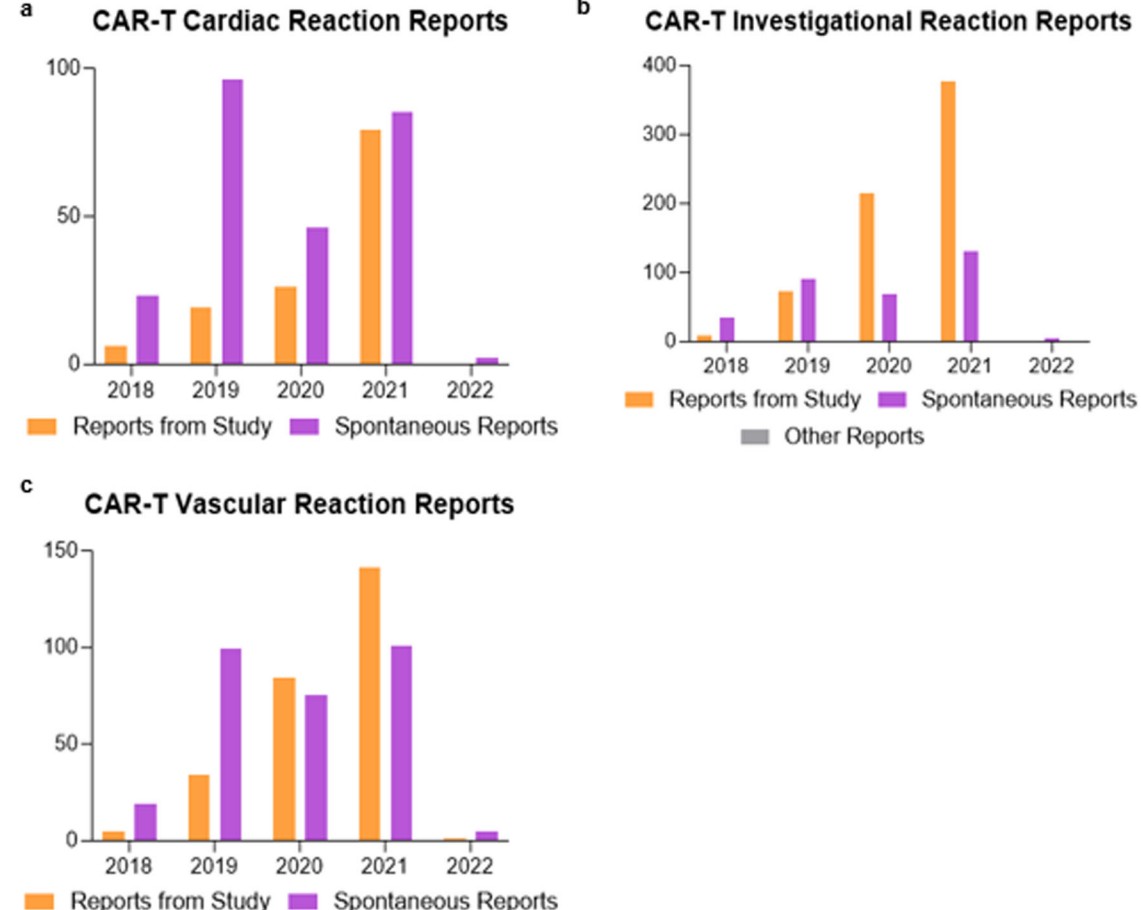

**Fig. 2 | Distribution of Individual Case Safety Reports (ICSRs) by report type.** ICSRs for CAR-T therapy were categorized by report type documented in VigiBase for (**a**) cardiac, (**b**) investigational, and (**c**) vascular adverse reactions from 2018 to 2022.

**Table 1 | Information components (IC) calculated for CAR-T therapy ($N_D$ = 5,576) for the given adverse reactions**

| Reaction Name | $N_A$ | $N_{AD}$ | IC | $IC_{025}$ |
|---|---|---|---|---|
| **Inflammatory/Effusive Events** | | | | |
| **C-Reactive Protein Increased** | **25,169** | **70** | **3.769** | **3.372** |
| **Capillary Leak Syndrome** | **1206** | **14** | **4.324** | **3.421** |
| Myocarditis | 28,484 | 2 | −1.211 | −3.804 |
| Pericarditis | 21,552 | 3 | −0.363 | −2.432 |
| **Pericardial Effusion** | **17,867** | **18** | **2.277** | **1.485** |
| Cardiac Tamponade | 2,897 | 0 | −1.053 | −11.377 |
| **Arrhythmic Events** | | | | |
| **Tachycardia** | **202,122** | **174** | **2.198** | **1.948** |
| **Atrial Fibrillation** | **79,948** | **45** | **1.569** | **1.073** |
| **Atrial Flutter** | **6773** | **8** | **2.274** | **1.062** |
| Electrocardiogram QT Prolonged | 27,465 | 9 | 0.763 | −0.376 |
| **Ventricular Tachycardia/ Ventricular Fibrillation** | **27,573** | **15** | **1.464** | **0.593** |
| **Cardiac Arrest** | **75,180** | **31** | **1.124** | **0.524** |
| **Hemodynamic Events** | | | | |
| **Hypotension** | **268,722** | **469** | **3.220** | **3.068** |
| **Sinus Tachycardia** | **14,075** | **18** | **2.571** | **1.779** |
| **Cardiomyopathy-Related Events** | | | | |
| **Stress Cardiomyopathy** | **2700** | **6** | **2.699** | **1.284** |
| **Ejection Fraction Decreased** | **10,708** | **11** | **2.209** | **1.184** |
| Cardiac Failure | 130,976 | 30 | 0.241 | −0.369 |
| **Cardiogenic Shock** | **7138** | **5** | **1.592** | **0.029** |

$IC_{025}$ > 0 is an indicator for significant adverse reaction signals, which are highlighted in bold; $N_D$ number of reports containing a given medication in VigiBase, $N_A$ number of reports containing a given adverse reaction in VigiBase, $N_{AD}$ number of reports containing a given drug-adverse reaction pair in VigiBase.

vicleucel, and n = 1 for ciltacabtagene autoleucel. A flowchart describing this process is illustrated in Fig. 1.

Given the heterogeneity of ICSRs reported to VigiBase, the ICSRs were categorized based on whether reporting was spontaneous (i.e., voluntary) or derived from clinical studies (Fig. 2). Of the 382 CAR-T ICSRs for all cardiac reactions, 130 ICSRs (34.0%) were reports from clinical study, while 252 reports (66.0%) were spontaneous. In contrast, 672 (67.0%) of the 1,003 CAR-T ICSRs for all investigational reactions were reports from clinical study, while 330 ICSRs (32.9%) were spontaneous with 1 report (0.1%) marked as other. For the 564 CAR-T ICSRs for all vascular reactions, 265 (47.0%) were reports from clinical study and 299 ICSRs (53.0%) were spontaneous.

## CAR-T therapy is associated with arrhythmic and systemic inflammatory adverse reactions

Information components (ICs) were calculated for CAR-T therapy for a series of adverse cardiovascular and inflammatory reactions, with $IC_{025}$ > 0 suggesting a statistically significant reaction signal. Apart from QT prolongation, CAR-T therapy was associated with most arrhythmic events, including atrial fibrillation and cardiac arrest (Table 1). CAR-T therapy was also associated with cardiomyopathy-related events, including stress cardiomyopathy, decreased ejection fraction, and cardiogenic shock. While no reports of cardiac tamponade were observed in VigiBase, pericardial effusion, increased C-reactive protein, capillary leak syndrome, hypotension, and sinus tachycardia were significant signals for CAR-T therapy. Of the six

CAR-T therapies studied, only tisagenlecleucel, axicabtagene ciloleucel, and brexucabtagene autoleucel had significant signals for the investigated cardiovascular adverse reactions (Supplementary Tables 3–8).

The ICSRs for each of the 13 significant reactions for CAR-T therapy were filtered by the cancer indication that necessitated CAR-T treatment. The three most reported indications were diffuse large B-cell lymphoma (DLBCL; N = 391), acute lymphocytic leukemia (ALL; N = 289), and mantle cell lymphoma (MCL; N = 21). DLBCL was the most reported indication for ICSRs containing CAR-T therapy and adverse reactions related to arrhythmias and cardiomyopathies, while ALL was the most reported indication for pericardial effusion and capillary leak syndrome (Table 2).

## Higher fatality rates of hypotension observed in tisagenlecleucel compared to axicabtagene ciloleucel in patients with diffuse large B-cell lymphoma

Tisagenlecleucel and axicabtagene ciloleucel are indicated for the treatment of relapsed/refractory DLBCL[27–29]. In VigiBase, there were 17,479 reports containing the DLBCL indication, which were filtered to 237 clinical study reports for tisagenlecleucel and 42 clinical study reports for axicabtagene ciloleucel for all cardiovascular and investigational adverse reactions. Reports from clinical study were used to mitigate the bias involved from voluntary reporting of severe adverse events. Tisagenlecleucel and axicabtagene ciloleucel shared eight significant reactions in common, which included increased C-reactive protein, pericardial effusion, tachycardia, atrial fibrillation, cardiac arrest, hypotension, sinus tachycardia, and decreased ejection fraction (Supplementary Tables 3, 4). Tisagenlecleucel was uniquely associated with capillary leak syndrome and cardiac failure, while axicabtagene ciloleucel was uniquely associated with atrial flutter, ventricular tachycardia/ventricular fibrillation, and stress cardiomyopathy.

Although axicabtagene ciloleucel had significantly greater reporting odds ratios (RORs) for hypotension than tisagenlecleucel in DLBCL patients (ROR: 2.54; 95% CI: 1.28–5.03; p = 0.012; $N_{AD}$ (axicabtagene ciloleucel) = 18, $N_{AD}$ (tisagenlecleucel) = 54), tisagenlecleucel had a significantly higher fatality rate for hypotension (50.0%) than axicabtagene ciloleucel (5.6%, p < 0.001) (Fig. 3 and Table 3). Additionally, tisagenlecleucel was associated with cardiac failure ($IC_{025}$ = 0.366), with an 85.7% fatality rate. Axicabtagene ciloleucel and tisagenlecleucel had similar reporting for atrial fibrillation in DLBCL patients (ROR: 2.20; 95% CI: 0.56–8.66; p = 0.22; $N_{AD}$ (axicabtagene ciloleucel) = 3, $N_{AD}$ (tisagenlecleucel) = 8), with no significant difference in fatality rates [50.0% (tisagenlecleucel) vs 33.3% (axicabtagene ciloleucel); p = 1.00]. Of the 11 DLBCL CAR-T patients reporting atrial fibrillation, 7 patients (63.6%) were treated with cyclophosphamide/fludarabine combination (Supplementary Table 9).

## Tisagenlecleucel is associated with fatal cardiac arrest and cardiac failure in pediatric acute lymphocytic leukemia

Tisagenlecleucel, brexucabtagene autoleucel, and blinatumomab are indicated for the treatment of relapsed/refractory ALL[30–32]. There were 28,803 reports in VigiBase containing the ALL indication, which were filtered to 354 clinical study reports for tisagenlecleucel, 230 clinical study reports for blinatumomab, and 2 clinical study reports for brexucabtagene autoleucel ICSRs for all cardiovascular and investigational adverse reactions. Blinatumomab shared six significant reactions in common with tisagenlecleucel, which included increased C-reactive protein, capillary leak syndrome, pericardial effusion, tachycardia, hypotension, and sinus tachycardia (Table 4 and Supplementary Table 3). Brexucabtagene autoleucel was associated with tachycardia and hypotension (Supplementary Table 5).

Tisagenlecleucel is indicated for the treatment of relapsed/refractory ALL in pediatric patients[30]. Blinatumomab, while commonly used for treating relapsed/refractory ALL in adult patients, is also indicated for treatment of pediatric relapsed/refractory ALL[33,34]. Consequently, safety reports from clinical studies were compared between tisagenlecleucel and blinatumomab for patients aged 17 or younger, resulting in 219 safety reports for tisagenlecleucel and 28 safety reports for blinatumomab for all cardiovascular and investigational adverse reactions.

**Table 2 | Distribution of CAR-T therapy-treated individual case safety reports (ICSRs) by cancer indication for each statistically significant adverse reaction signal**

| Reaction Name | Number of Reports for Each Indication | | | | | |
|---|---|---|---|---|---|---|
| | ALL | DLBCL | MCL | Other Lymphomas and Leukemias | Unspecified Lymphomas and Leukemias | Unknown Indication |
| Inflammatory/Effusive Events | | | | | | |
| C-Reactive Protein Increased | 26 | 30 | 1 | 1* | 7 | 5 |
| Capillary Leak Syndrome | 7 | 5 | 0 | 0 | 0 | 2 |
| Pericardial Effusion | 9 | 4 | 0 | 1 | 2 | 2 |
| Arrhythmic Events | | | | | | |
| Tachycardia | 49 | 93 | 6 | 4 | 11 | 11 |
| Atrial Fibrillation | 0 | 31 | 4 | 0 | 4 | 6 |
| Atrial Flutter | 0 | 5 | 2 | 1 | 0 | 0 |
| VT/VF | 5 | 9 | 1 | 0 | 0 | 0 |
| Cardiac Arrest | 7 | 19 | 0 | 1 | 1 | 3 |
| Hemodynamic Events | | | | | | |
| Hypotension | 176 | 176 | 7 | 19 | 48 | 43 |
| Sinus Tachycardia | 8 | 5 | 0 | 3 | 2 | 0 |
| Cardiomyopathy-Related Events | | | | | | |
| Stress Cardiomyopathy | 0 | 5 | 0 | 0 | 1 | 0 |
| Ejection Fraction Decreased | 2 | 6 | 0 | 1 | 2 | 0 |
| Cardiogenic Shock | 0 | 3 | 0 | 1 | 0 | 1 |

*ALL* Acute lymphocytic leukemia, *DLBCL* diffuse large B-cell lymphoma, *MCL* mantle cell lymphoma, *VT/VF* ventricular tachycardia/ventricular fibrillation. The asterisk (*) denotes the one patient who had both acute lymphocytic leukemia and diffuse large B-cell lymphoma for the adverse reaction of increased C-reactive protein.

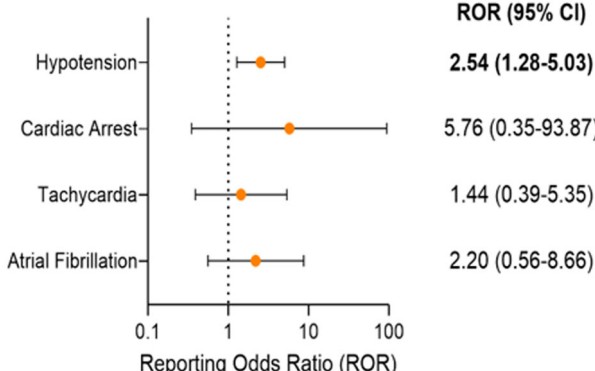

**Fig. 3 | Comparison of CAR-T Cardiotoxicities for Diffuse Large B-Cell Lymphoma (DLBCL).** Reporting odds ratios were compared for significant adverse reactions between tisagenlecleucel (N = 237 reports) and axicabtagene ciloleucel (N = 42 reports) for DLBCL patients. Significant adverse reactions include atrial fibrillation (n = 8 reports [tisagenlecleucel] and n = 3 reports [axicabtagene ciloleucel]), tachycardia (n = 12 reports [tisagenlecleucel] and n = 3 reports [axicabtagene ciloleucel]), cardiac arrest (n = 1 reports [tisagenlecleucel] and n = 1 reports [axicabtagene ciloleucel]), and hypotension (n = 54 reports [tisagenlecleucel] and n = 18 reports [axicabtagene ciloleucel]). Error bars denote 95% confidence intervals.

While blinatumomab had significantly higher reporting for pericardial effusion compared to tisagenlecleucel (ROR: 8.64; 95% CI: 1.65–45.12; p = 0.021; $N_{AD}$ (tisagenlecleucel) = 3, $N_{AD}$ (blinatumomab) = 3), the fatality rates were not significantly different [100% (tisagenlecleucel) vs 66.7% (blinatumomab); p = 1.00] (Fig. 4 and Table 5). Additionally, tisagenlecleucel-associated cardiac arrest and cardiac failure had high fatality rates of 100% and 80.0% respectively. Blinatumomab and

tisagenlecleucel had similar reporting for hypotension (ROR: 0.44; 95% CI: 0.16–1.22; p = 0.13; $N_{AD}$ (tisagenlecleucel) = 72, $N_{AD}$ (blinatumomab) = 5) as well as similar fatality rates [34.7% (tisagenlecleucel) vs 20.0% (blinatumomab); p = 0.66] in pediatric ALL patients. Similarly, tisagenlecleucel and blinatumomab had comparable fatality rates for tachycardia [47.1% (tisagenlecleucel) vs 50.0% (blinatumomab); p = 1.00]. Of the 72 pediatric ALL patients treated with tisagenlecleucel who reported having hypotension, about 83.3% were treated with cyclophosphamide/fludarabine combination (Supplementary Data 1).

## Discussion

CAR-T therapy has found applications in the treatment of numerous cancers, including ALL, DLBCL, MCL, and multiple myeloma[35–37]. Indeed, our study highlighted that DLBCL, ALL, and MCL were the three most reported indications for CAR-T therapy in VigiBase, and the two multiple myeloma CAR-T reports for cardiogenic shock and hypotension corresponded to the medications of ciltacabtagene autoleucel and idecabtagene vicleucel, respectively. Unfortunately, the immense benefits of CAR-T therapy might be dampened by the serious drawbacks resulting from cytokine release syndrome, which can present a complex cardiovascular toxicity profile[17]. Information component analyses in our study showed that CAR-T therapy was associated with serious arrhythmias, including atrial fibrillation, ventricular tachycardia/ventricular fibrillation, and cardiac arrest, as well as cardiomyopathy-related events, including decreased ejection fraction and stress cardiomyopathy. As such, our study, which was based on safety data from six CAR-T cell products, provides further support to recent studies that investigated the cardiovascular toxicity profiles of CAR-T therapy[38–41].

A key highlight of our study is the cancer-specific comparison of cardiovascular toxicity profiles between CAR-T cell products, as well as between CAR-T therapy and bispecific T-cell engagers. Salem et al. and Goldman et al. performed pharmacovigilance analyses using VigiBase and the Food and Drug Administration (FDA) Adverse Event Reporting System (FAERS) database to illustrate that tisagenlecleucel and axicabtagene

**Table 3 | Case-seriousness rates calculated for cardiovascular and inflammatory adverse events associated with the CAR-T therapies of tisagenlecleucel and axicabtagene ciloleucel for diffuse large B-cell lymphoma patients**

| Adverse Reaction | Total Cases | Number of Fatal Cases (% of Total) | Number of Life-Threatening Cases (% of Total) | Number of Caused/ Prolonged Hospitalization Cases (% of Total) |
|---|---|---|---|---|
| Tisagenlecleucel | | | | |
| Atrial fibrillation | 8 | 4 (50.0%) | 1 (12.5%) | 2 (25.0%) |
| Tachycardia | 12 | 11 (91.7%) | 0 (0%) | 0 (0%) |
| Sinus tachycardia | 1 | 0 (0%) | 0 (0%) | 0 (0%) |
| Cardiac arrest | 1 | 1 (100%) | 0 (0%) | 0 (0%) |
| Pericardial effusion | 2 | 1 (50.0%) | 0 (0%) | 1 (50.0%) |
| Cardiac failure | 7 | 6 (85.7%) | 0 (0%) | 1 (14.3%) |
| Hypotension | 54 | 27 (50.0%) | 2 (3.7%) | 7 (13.0%) |
| Capillary leak syndrome | 1 | 1 (100%) | 0 (0%) | 0 (0%) |
| Axicabtagene Ciloleucel | | | | |
| Atrial fibrillation | 3 | 1 (33.3%) | 0 (0%) | 1 (33.3%) |
| Atrial flutter | 1 | 0 (0%) | 1 (100%) | 0 (0%) |
| Tachycardia | 3 | 1 (33.3%) | 0 (0%) | 2 (66.7%) |
| Cardiac arrest | 1 | 1 (100%) | 0 (0%) | 0 (0%) |
| Hypotension | 18 | 1 (5.6%) | 4 (22.2%) | 9 (50.0%) |

ciloleucel were associated with pericardial disorders and tachyarrhythmias, with Goldman et al. further showing that axicabtagene ciloleucel was associated with venous thromboembolic events[38,39]. Tisagenlecleucel and axicabtagene ciloleucel are both CD19-targeted CAR-T cell products approved for the treatment of relapsed/refractory DLBCL[27–29], and hence comparisons between the two CAR-T cell products in our study were restricted specifically to the DLBCL patient population. Unlike the other CAR-T cell products and the overall CAR-T category, tisagenlecleucel was uniquely associated with cardiac failure, with an 85.7% fatality rate in DLBCL patients. Additionally, tisagenlecleucel had a significantly higher fatality rate of 50.0% for hypotension compared to 5.6% for axicabtagene ciloleucel, even though axicabtagene ciloleucel had significantly greater reporting for hypotension. As hypotension is a characteristic symptom for Grade 2 and higher CRS according to the ASTCT Consensus Grading[18], tisagenlecleucel seemed to be associated with a more severe and fatal form of CRS compared to axicabtagene ciloleucel in DLBCL patients, thereby potentially highlighting differences in CRS manifestations between CAR-T cell products.

CRS is characterized by increased levels of proinflammatory cytokines, including IL-6 and IFN-γ[15,18,42]. Based on the current understanding of CRS mechanisms, CAR-T cells first recognize tumor antigens and then release large amounts of cytokines, perforin and granzymes, leading to a significant release of gasdermin and resulting in tumor pyroptosis, which is a proinflammatory form of programmed cell death mediated by gasdermin[42–45]. Pyroptotic tumor cells can release large amounts of damage-associated molecular patterns (DAMPs), which include double-stranded DNA, adenosine triphosphate (ATP), and high-mobility group box 1 (HMGB1), resulting in the activation of macrophages[42,43,46]. ATP can lead to the formation of the NLRP3 inflammasome and maturation of caspase 1 in macrophages, with caspase 1 converting gasdermin D into its active form and leading to pyroptosis in macrophages[42,46]. Pyroptotic macrophages in turn produce large amounts of DAMPs, resulting in the activation of even more macrophages[42,46]. Meanwhile, HMGB1 binds to TLR2 and TLR4 found on surface of macrophages, which leads to adaptor proteins MyD88 and TRIF activating mitogen-activated protein kinases (MAPKs) and IκB kinase (IKK) that in turn regulate the release of cytokines, including IL-6[42,43,46]. The release of cytokines from macrophages and CAR-T cells can injure endothelial cells and contribute to edema, coagulopathy, and increased vascular permeability and leakage[42,47].

DAMPs play a similar role in the cardiotoxicity of immune checkpoint inhibitors (ICIs), whereby short-term ICI therapy is associated with a proinflammatory cytokine storm and increased myocardial expression of MyD88 and NLRP3, resulting in increased myocardial and vascular inflammation[11]. Mechanistically, a deficiency in PD-1/PD-L1 is linked to higher levels of pro-atherosclerotic cytokines, IFN-γ and TNF-α, with increased numbers of CD4+ and CD8+ T cells present in atherosclerotic lesions[48–50]. Similarly, blocking the CD28-CD80/86 co-stimulatory T-cell activation with abatacept, which is a CTLA-4Ig fusion protein, prevented CD4+ T cell activation and reduced the accelerated development of atherosclerosis in hypercholesterolemic ApoE3*Leiden mice[48,51]. However, these mechanistic studies are still in progress, and future studies are needed to investigate the interplay between the PD-1/PD-L1/CTLA-4 axis and the CRS associated with CAR-T therapy and pyroptosis.

Blinatumomab is a bispecific T-cell engager that binds to both CD3 found on T cells and CD19 found on B cells, thereby bringing T cells to malignant B cells and initiating a cytotoxic T-cell response[52]. Blinatumomab is approved for the treatment of relapsed/refractory ALL, along with tisagenlecleucel and brexucabtagene autoleucel, where both CAR-T cell products target CD19[30–32]. While information component analyses showed that brexucabtagene autoleucel was associated with hypotension and tachycardia, the small sample size of only two ALL reports limited direct comparisons with blinatumomab and tisagenlecleucel.

In pediatric ALL patients, blinatumomab had significantly higher reporting odds ratios for pericardial effusion compared to tisagenlecleucel, but the fatality rates for pericardial effusion were similar between the two groups. Tisagenlecleucel and blinatumomab also had similar fatality rates for tachycardia, sinus tachycardia, and hypotension. Unlike blinatumomab, tisagenlecleucel was associated with cardiac arrest and cardiac failure, with high fatality rates of 100% and 80.0% respectively. Recently, two new CD20/CD3 bispecific T-cell engagers, glofitamab and mosunetuzumab, were approved for the treatment of relapsed/refractory diffuse large B-cell lymphoma and follicular lymphoma respectively[53,54]. Our study comparing cardiovascular toxicity profiles between CD19-targeted CAR-T therapy and the CD19/CD3 bispecific T-cell engager, blinatumomab, can serve as a foundation for future studies that compare toxicity profiles between CAR-T therapy and bispecific T-cell engagers, where both therapies target the same tumor antigen, and uncover mechanisms that would explain differences in CRS presentation.

In both DLBCL CAR-T patients reporting atrial fibrillation and pediatric ALL CAR-T patients reporting hypotension, most of the patient populations (63.6% for DLBCL and 83.3% of ALL) were treated with the chemotherapy regimen of cyclophosphamide/fludarabine. The

cyclophosphamide/fludarabine combination is an effective conditioning regimen that can create a favorable environment for CAR-T cell expansion and function through mechanisms involving lymphodepletion and elimination of immunosuppressive cells[55,56]. However, cyclophosphamide is cardiotoxic, and so additional studies are needed to explore the roles that the

conditioning regimen might contribute to the cardiotoxicities associated with CAR-T therapy[57].

An important limitation of this study is that the data reported to VigiBase comes from various sources, so the probability that a suspected adverse reaction is related to a particular drug is not the same in all cases[23]. VigiBase safety reports are affected by biases and data heterogeneity, and there is no prevalence information concerning the total number of patients treated with the drug[23]. Considering the diversity in data sources, including both spontaneous reporting and trial data, it is essential to acknowledge that any observed variations in toxicity among different cell therapies could potentially be attributed to this factor. The problem of selection bias involved from the voluntary reporting of adverse events was mitigated by restricting the comparisons to reports from clinical studies. However, this approach led to a decrease in the sample size of reports available for comparisons, and information on time to onset of adverse reactions associated with a drug was limited. Furthermore, comparisons between blinatumomab

**Table 4 | Information components (IC) calculated for blinatumomab (N$_D$ = 4808) for the given adverse reactions**

| Reaction Name | N$_A$ | N$_{AD}$ | IC | IC$_{025}$ |
|---|---|---|---|---|
| Inflammatory/Effusive Events | | | | |
| **C-Reactive Protein Increased** | **25,169** | **20** | **2.179** | **1.428** |
| **Capillary Leak Syndrome** | **1,206** | **8** | **3.616** | **2.404** |
| Myocarditis | 28,484 | 0 | −3.339 | −13.663 |
| Pericarditis | 21,552 | 1 | −1.397 | −5.180 |
| **Pericardial Effusion** | **17,867** | **11** | **1.775** | **0.751** |
| Cardiac Tamponade | 2,897 | 1 | 0.638 | −3.145 |
| Arrhythmic Events | | | | |
| **Tachycardia** | **202,122** | **61** | **0.905** | **0.480** |
| Atrial Fibrillation | 79,948 | 14 | 0.125 | −0.778 |
| Atrial Flutter | 6,773 | 2 | 0.658 | −1.935 |
| Electrocardiogram QT Prolonged | 27,465 | 5 | 0.168 | −1.394 |
| Ventricular Tachycardia/ Ventricular Fibrillation | 27,573 | 3 | −0.489 | −2.559 |
| Cardiac Arrest | 75,180 | 7 | −0.741 | −2.043 |
| Hemodynamic Events | | | | |
| **Hypotension** | **268,722** | **96** | **1.149** | **0.811** |
| **Sinus Tachycardia** | **14,075** | **7** | **1.446** | **0.144** |
| Cardiomyopathy-Related Events | | | | |
| Stress Cardiomyopathy | 2,700 | 0 | −0.899 | −11.222 |
| Ejection Fraction Decreased | 10,708 | 2 | 0.175 | −2.418 |
| Cardiac Failure | 130,976 | 9 | −1.176 | −2.315 |
| Cardiogenic Shock | 7,138 | 2 | 0.606 | −1.987 |

IC$_{025}$ > 0 is an indicator for significant adverse reaction signals, which are highlighted in bold; N$_D$ number of reports containing a given medication in VigiBase; N$_A$ number of reports containing a given adverse reaction in VigiBase; N$_{AD}$ number of reports containing a given drug-adverse reaction pair in VigiBase.

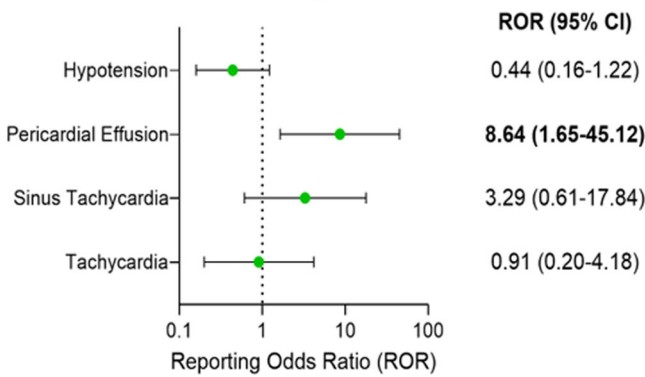

**Fig. 4 | Cardiovascular toxicity comparisons for immunotherapies targeting Acute Lymphocytic Leukemia (ALL).** Reporting odds ratios were compared for significant adverse reactions between tisagenlecleucel (N = 219 reports) and blinatumomab (N = 28 reports) for pediatric ALL patients (aged 17 or younger). Significant adverse reactions include tachycardia (n = 17 reports [tisagenlecleucel] and n = 2 reports [blinatumomab]), sinus tachycardia (n = 5 reports [tisagenlecleucel] and n = 2 reports [blinatumomab]), pericardial effusion (n = 3 reports [tisagenlecleucel] and n = 3 reports [blinatumomab]), and hypotension (n = 72 reports [tisagenlecleucel] and n = 5 reports [blinatumomab]). Error bars denote 95% confidence intervals.

**Table 5 | Case-seriousness rates calculated for cardiovascular and inflammatory adverse events associated with tisagenlecleucel and blinatumomab for acute lymphocytic leukemia patients (aged 17 years or younger in both drug categories)**

| Adverse Reaction | Total Cases | Number of Fatal Cases (% of Total) | Number of Life-Threatening Cases (% of Total) | Number of Caused/ Prolonged Hospitalization Cases (% of Total) |
|---|---|---|---|---|
| Tisagenlecleucel | | | | |
| Tachycardia | 17 | 8 (47.1%) | 4 (23.5%) | 3 (17.6%) |
| Sinus tachycardia | 5 | 2 (40.0%) | 3 (60.0%) | 0 (0%) |
| Cardiac arrest | 3 | 3 (100%) | 0 (0%) | 0 (0%) |
| Pericardial effusion | 3 | 3 (100%) | 0 (0%) | 0 (0%) |
| Cardiac failure | 5 | 4 (80.0%) | 0 (0%) | 0 (0%) |
| Hypotension | 72 | 25 (34.7%) | 20 (27.8%) | 5 (6.9%) |
| Capillary leak syndrome | 1 | 1 (100%) | 0 (0%) | 0 (0%) |
| Blinatumomab | | | | |
| Tachycardia | 2 | 1 (50.0%) | 1 (50.0%) | 0 (0%) |
| Sinus tachycardia | 2 | 2 (100%) | 0 (0%) | 0 (0%) |
| Pericardial effusion | 3 | 2 (66.7%) | 1 (33.3%) | 0 (0%) |
| Hypotension | 5 | 1 (20.0%) | 2 (40.0%) | 2 (40.0%) |

and tisagenlecleucel were made for patients aged 17 or younger, as VigiBase included demographic reports for the 18-44 age group. Given the retrospective nature of VigiBase, the calculated information components and reporting odds ratios could only be used to show an association, not a causation, between a reaction and a drug category. Nonetheless, the analyses from VigiBase can be used to develop hypotheses for future studies that further investigate the cardiovascular profile of CRS seen in CAR-T therapy.

## Conclusion

In this comprehensive pharmacovigilance analysis, tisagenlecleucel was associated with significantly higher fatality rates for hypotension compared to axicabtagene ciloleucel in DLBCL patients but had similar fatality rates compared to blinatumomab in pediatric ALL patients. Tisagenlecleucel was also associated with cardiac failure with high fatality rates in both DLBCL and pediatric ALL patients. Identification of such risk factors may help with efforts to mitigate future risk, and future studies are needed to further examine the mechanisms surrounding the cardiovascular safety profiles of CAR-T therapy and bispecific T-cell engagers. We conclude that these therapies, despite their potential life-saving benefits, are associated with a significant risk of cardiac toxicity. To ensure proper management, it is crucial to engage experienced physicians with expertise in interdisciplinary approaches, including skilled cardiologists or cardio-oncologists, particularly in managing these toxicities.

## Data availability

The data that support the findings of this study include publicly available datasets from VigiBase, World Health Organization (WHO) global database. While these datasets are publicly accessible, some data may have access restrictions. Interested parties can access the data through the WHO Global Health Observatory data repository at [https://who-umc.org/vigibase/], and additional access details can be requested from the corresponding author upon written request. Source data for the figures are available as Supplementary Data 2.

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

## Acknowledgements

MedDRA® trademark is registered by the International Council for Harmonization of Technical Requirements for Pharmaceuticals for Human Use (ICH). Moreover, the information gathered from VigiBase does not represent the opinion of the Uppsala Monitoring Centre (UMC) or the World Health Organization. SP received support from NIH/NHLBI (grant no. R01HL150266). UCS received support from NIH/NHLBI (grant no. R01HL152090).

## Author contributions

B.K., S.S.S., I.P., S.H.O., S.P., and U.C.S. participated in conceptualization of the research ideas, retrieval of the clinical datasets, data analysis and manuscript writing. U.C.S. was also responsible for overall study design, mentorship, manuscript preparation and funding this study.

## Competing interests

The authors declare no competing interests.
