## [Peer Review File · Communications Medicine]

Reviewers' comments:

Reviewer #1 (Remarks to the Author):

Karthikeyan et al. conducted a retrospective analysis of adverse effects induced by CAR T cells (tisagenlecleucel and axicabtagene ciloleucel) and Blinatumomab, bispecific T-cell engagers, which are clinically indicated for the treatment of relapsed/refractory diffuse large B cell lymphoma (DLBCL) and acute lymphocytic leukemia (ALL). Although these therapies rely on a similar mechanism of action, as both promote the killing of tumor B cells by T cells through recognition of the B cell surface antigen B4 (CD19), their adverse effects on cardiac function differ in severity and type. In patients with DLBCL, tisagenlecleucel increases the risk of mortality associated with hypotension and tachycardia and heart failure compared to axicabtagene ciloleucel, which is associated with a greater risk of ventricular tachycardia/ventricular fibrillation. In ALL patients, treatment with blinatumomab has hypertriglyceridemia as its main adverse event, while the risk of hypotension and tachycardia is reduced compared to tisagenlecleucel. The authors note that this statistical analysis of adverse effects associated with CART and bispecific antibodies could have clinical implications, providing insight into the cardiovascular risks associated with these therapies and the possibility of reducing them based on a patient's medical history. However, the study does not determine the percentage of adverse cases compared to the total number of patients treated, and the analysis may underestimate adverse events due to the inclusion of both clinical studies and spontaneous patient reports. The reporting of adverse events in the latter case may be subjective and dependent on patients' sensitivity, leading to both over- and underestimation of cases. Despite these limitations, the study was well-designed, and the article was well-structured. Overall, I have no major concerns about the study.

Reviewer #2 (Remarks to the Author):

Manuscript titled “Cardiotoxic Profiles of CAR-T Therapy and Bispecific T-Cell Engagers in Hematological Cancers “ is a very interesting article in the field of cardioncology. The overall structure is of good quality and easy to read. Methods and results are clear and results corroborate the initial hypothesis of the authors. Figures and tables are of sufficient quality and easy to read as well as to understand to readers.

However, manuscript need some improvements, specifically in introduction and/or discussion. Here the points:

1. In introduction, authors should describe the key role of Myd-88 and NLRP-3 pathways in anticancer functions of immunotherapy, including immune checkpoint inhibitors and CAR-T and

their role also in vascular and myocardial affections, including myocarditis and vasculitis (cite doi: 10.3389/fcvm.2022.930797)

2. ICIs therapy increases RR of atherosclerosis in cancer patients through the involvement of immune-derived inflammation in vascular tissue. Please discuss this point and differences with CAR-T therapy.

Based on these changes, the article could be suitable for publication in this journal.

Reviewer #3 (Remarks to the Author):

I have gladly reviewed the manuscript by Karthikeyan et al on the cardiotoxicity of CART-cell products and T-cell engagers in patients with hematological malignancies. The study is based on a WHO-based pharmacovigilance registry (VigiBase), with its many potential biases.

I have a number of comments:

MAJOR:

- The major problem of this study is the bias related to the voluntary decision of a physician to report an adverse or not, and the logical higher probability of reporting severe vs. mild adverse events. In this regard, the "severity" calculations are, by definition biased. One of them particularly caught my attention, which is that hypertriglyceridemia was life-threatening in all patients treated with blinatumomab. This is somehow surprising because hypertriglyceridemia is usually not very relevant as an adverse event and, when it is relevant, it is in the context of HLH-type of complication, which is far more frequent and dangerous in patients receiving CART-cells.

- I am also shocked to see that not all patients receiving CART-cells also received FluCy chemotherapy. Most if not all clinical trials and protocols involving CART-cells generally include FluCy lymphodepleting chemotherapy, which is not the case for bispecific.

- Comparing blina vs. tisa-cel is not really fair because the approved indication for tisa-cel is pediatric ALL and blina is mostly used in adult patients.

MINOR:

- ICSR is not spelled out until much later, probably because the order of the paper was different in a previous submission (Methods before Results).

- The manuscript is written in such a way that all the knowledge (all the "truths") seem to come from the tables and the figures. "From table X we conclude that..." "Table X showed that tisa-cel had a significantly higher fatality rate". I would somehow recommend to rephrase most of these sentences.

We greatly appreciate reviewers' detailed evaluation and constructive feedback of our manuscript. Based on their feedback, we updated our comparisons to stick strictly to Vigibase safety reports obtained from clinical studies to mitigate the bias involved from the voluntary reporting of adverse events. We have carefully addressed specific aspects of each reviewer's comments, and revised the introduction, results, and discussion reflecting these changes, while also updating study limitations.

Reviewer #1 (Remarks to the Author):

Comment: Karthikeyan et al. conducted a retrospective analysis of adverse effects induced by CAR T cells (tisagenlecleucel and axicabtagene ciloleucel) and Blinatumomab, bispecific T-cell engagers, which are clinically indicated for the treatment of relapsed/refractory diffuse large B cell lymphoma (DLBCL) and acute lymphocytic leukemia (ALL). Although these therapies rely on a similar mechanism of action, as both promote the killing of tumor B cells by T cells through recognition of the B cell surface antigen B4 (CD19), their adverse effects on cardiac function differ in severity and type. In patients with DLBCL, tisagenlecleucel increases the risk of mortality associated with hypotension and tachycardia and heart failure compared to axicabtagene ciloleucel, which is associated with a greater risk of ventricular tachycardia/ventricular fibrillation. In ALL patients, treatment with blinatumomab has hypertriglyceridemia as its main adverse event, while the risk of hypotension and tachycardia is reduced compared to tisagenlecleucel. The authors note that this statistical analysis of adverse effects associated with CART and bispecific antibodies could have clinical implications, providing insight into the cardiovascular risks associated with these therapies and the possibility of reducing them based on a patient's medical history.

Response: We greatly thank the reviewer for the positive feedback and agree with the reviewer's point on the limitations of the study. Unfortunately, Vigibase does not provide information about the total number of patients treated, which makes it difficult to calculate the percentage of adverse cases. This is reflected in the limitations below:

Vigibase safety reports are affected by biases and data heterogeneity, and there is no prevalence information concerning the total number of patients treated with the drug¹.

Reference:

1. Lindquist, M. Vigibase, the WHO Global ICSR Database System: Basic Facts. Drug information journal : DIJ / Drug Information Association 42, 409-419, doi:10.1177/009286150804200501 (2008).

Comment: However, the study does not determine the percentage of adverse cases compared to the total number of patients treated, and the analysis may underestimate adverse events due to the inclusion of both clinical studies and spontaneous patient reports. The reporting of adverse events in the latter case may be subjective and dependent on patients' sensitivity, leading to both over- and underestimation of cases.

Response: We agree with the reviewer that including the spontaneous patient reports might result in bias concerning subjectivity involved in reporting adverse events. We therefore revised the analysis to stick strictly to safety reports from clinical studies to mitigate this bias. These changes are reflected in the revised manuscript, with the following takeaways from the Results:

Higher Fatality Rates of Hypotension Observed in Tisagenlecleucel Compared to Axicabtagene Ciloleucel in Patients with Diffuse Large B-Cell Lymphoma

In VigiBase, there were 17,479 reports containing the DLBCL indication, which were filtered to 237 clinical study reports for tisagenlecleucel and 42 clinical study reports for axicabtagene ciloleucel for all cardiovascular and investigational adverse reactions. Reports from clinical study were used to mitigate the bias involved from voluntary reporting of severe adverse events.

Although axicabtagene ciloleucel had significantly greater reporting odds ratios (RORs) for hypotension than tisagenlecleucel in DLBCL patients (ROR: 2.54; 95% CI: 1.28-5.03; $p = 0.012$; N_{AD} (axicabtagene ciloleucel) = 18, N_{AD} (tisagenlecleucel) = 54), tisagenlecleucel had a significantly higher fatality rate for hypotension (50.0%) than axicabtagene ciloleucel (5.6%, $p < 0.001$) (Figure 3 and Table 3). Additionally, tisagenlecleucel was associated with cardiac failure ($IC_{025} = 0.366$), with an 85.7% fatality rate. Axicabtagene ciloleucel and tisagenlecleucel had similar reporting for atrial fibrillation in DLBCL patients (ROR: 2.20; 95% CI: 0.56-8.66; $p = 0.22$; N_{AD} (axicabtagene ciloleucel) = 3, N_{AD} (tisagenlecleucel) = 8), with no significant difference in fatality rates [50.0% (tisagenlecleucel) vs 33.3% (axicabtagene ciloleucel); $p = 1.00$]. Of the 11 DLBCL CAR-T patients reporting atrial fibrillation, 7 patients (63.6%) were treated with cyclophosphamide/fludarabine combination (Supplementary Table 9).

Tisagenlecleucel is Associated with Fatal Cardiac Arrest and Cardiac Failure in Pediatric Acute Lymphocytic Leukemia

There were 28,803 reports in VigiBase containing the ALL indication, which were filtered to 354 clinical study reports for tisagenlecleucel, 230 clinical study reports for blinatumomab, and 2 clinical study reports for brexucabtagene autoleucel ICSRs for all cardiovascular and investigational adverse reactions.

Tisagenlecleucel is indicated for the treatment of relapsed/refractory ALL in pediatric patients. Blinatumomab, while commonly used for treating relapsed/refractory ALL in adult patients, is also indicated for treatment of pediatric relapsed/refractory ALL. Consequently, safety reports from clinical studies were compared between tisagenlecleucel and blinatumomab for patients aged 17 or younger, resulting in 219 safety reports for tisagenlecleucel and 28 safety reports for blinatumomab for all cardiovascular and investigational adverse reactions.

While blinatumomab had significantly higher reporting for pericardial effusion compared to tisagenlecleucel (ROR: 8.64; 95% CI: 1.65-45.12; $p = 0.021$; N_{AD} (tisagenlecleucel) = 3, N_{AD} (blinatumomab) = 3), the fatality rates were not significantly different [100% (tisagenlecleucel) vs 66.7% (blinatumomab); $p = 1.00$] (Figure 4 and Table 5). Additionally, tisagenlecleucel-associated cardiac arrest and cardiac failure had high fatality rates of 100% and 80.0% respectively. Blinatumomab and tisagenlecleucel had similar reporting for hypotension (ROR: 0.44; 95% CI: 0.16-1.22; $p = 0.13$; N_{AD} (tisagenlecleucel) = 72, N_{AD} (blinatumomab) = 5) as well as similar fatality rates [34.7% (tisagenlecleucel) vs 20.0% (blinatumomab); $p = 0.66$] in pediatric ALL patients. Similarly, tisagenlecleucel and blinatumomab had comparable fatality rates for tachycardia [47.1% (tisagenlecleucel) vs 50.0% (blinatumomab); $p = 1.00$]. Of the 72 pediatric ALL patients treated with tisagenlecleucel who reported having hypotension, about 83.3% were treated with cyclophosphamide/fludarabine combination (Supplementary Table 10).

Reviewer #2 (Remarks to the Author):

Manuscript titled “Cardiotoxic Profiles of CAR-T Therapy and Bispecific T-Cell Engagers in Hematological Cancers “ is a very interesting article in the field of cardioncology. The overall structure is of good quality and easy to read. Methods and results are clear and results corroborate the initial hypothesis of the authors. Figures and tables are of sufficient quality and easy to read as well as to understand to readers.

However, manuscript need some improvements, specifically in introduction and/or discussion. Here the points:

Comment 1. *In introduction, authors should describe the key role of Myd-88 and NLRP-3 pathways in anticancer functions of immunotherapy, including immune checkpoint inhibitors and CAR-T and their role also in vascular and myocardial affections, including myocarditis and vasculitis (cite doi: 10.3389/fcvm.2022.930797)*

Response: We greatly thank the reviewer for providing a very important point concerning the Myd-88 and NLRP-3 pathways. We updated the Introduction to reflect this as shown below:

Immune checkpoint inhibitors (ICIs) that block PD-1 can help prevent CAR-T cell exhaustion and thus improve the effectiveness of CAR-T therapy ¹. However, ICIs are cardiotoxic ², whereby even short-term treatment of ICIs could result in a proinflammatory cytokine storm and increased production of damage-associated molecular patterns (DAMPs) in myocardial tissues, with increased myocardial expression of MyD88 and NLRP3 promoting fibrosis, hypertrophy, and myocardial inflammation ³.

References:

1. Grosser, R., Cherkassky, L., Chintala, N. & Adusumilli, P. S. Combination Immunotherapy with CAR T Cells and Checkpoint Blockade for the Treatment of Solid Tumors. *Cancer Cell* 36, 471-482, doi:10.1016/j.ccell.2019.09.006 (2019).
2. Xu, S., Sharma, U. C., Tuttle, C. & Pokharel, S. Immune Checkpoint Inhibitors: Cardiotoxicity in Pre-clinical Models and Clinical Studies. *Front Cardiovasc Med* 8, 619650, doi:10.3389/fcvm.2021.619650 (2021).
3. Quagliariello, V. et al. Immune checkpoint inhibitor therapy increases systemic SDF-1, cardiac DAMPs Fibronectin-EDA, S100/Calgranulin, galectine-3, and NLRP3-MyD88-chemokine pathways. *Front Cardiovasc Med* 9, 930797, doi:10.3389/fcvm.2022.930797 (2022).

Comment 2. *ICIs therapy increases RR of atherosclerosis in cancer patients through the involvement of immune-derived inflammation in vascular tissue. Please discuss this point and differences with CAR-T therapy.*

Based on these changes, the article could be suitable for publication in this journal.

Response: We thank the reviewer for highlighting this important point, and we revised the Discussion to elaborate more about the mechanisms surrounding cytokine release syndrome associated with CAR-T therapy and how that is related to the ICI-associated atherosclerosis.

CRS is characterized by increased levels of proinflammatory cytokines, including IL-6 and IFN- γ ^{1,2,3}. Based on the current understanding of CRS mechanisms, CAR-T cells first recognize tumor antigens and then release large amounts of cytokines, perforin and granzymes, leading to a significant release of gasdermin and resulting in tumor pyroptosis, which is a proinflammatory form of programmed cell death mediated by gasdermin³⁻⁶. Pyroptotic tumor cells can release large amounts of damage-associated molecular patterns (DAMPs), which include double-stranded DNA, adenosine triphosphate (ATP), and high-mobility group box 1 (HMGB1), resulting in the activation of macrophages^{3,4,7}. ATP can lead to the formation of the NLRP3 inflammasome and maturation of caspase 1 in macrophages, with caspase 1 converting gasdermin D into its active form and leading to pyroptosis in macrophages^{3,7}. Pyroptotic macrophages in turn produce large amounts of DAMPs, resulting in the activation of even more macrophages^{3,7}. Meanwhile, HMGB1 binds to TLR2 and TLR4 found on surface of macrophages, which leads to adaptor proteins MyD88 and TRIF activating mitogen-activated protein kinases (MAPKs) and I κ B kinase (IKK) that in turn regulate the release of cytokines, including IL-6^{3,4,7}. The release of cytokines from macrophages and CAR-T cells can injure endothelial cells and contribute to edema, coagulopathy, and increased vascular permeability and leakage^{3,8}.

DAMPs play a similar role in the cardiotoxicity of immune checkpoint inhibitors (ICIs), whereby short-term ICI therapy is associated with a proinflammatory cytokine storm and increased myocardial expression of MyD88 and NLRP3, resulting in increased myocardial and vascular inflammation⁹. Mechanistically, a deficiency in PD-1/PD-L1 is linked to higher levels of pro-atherosclerotic cytokines, IFN- γ and TNF- α , with increased numbers of CD4+ and CD8+ T cells present in atherosclerotic lesions¹⁰⁻¹². Similarly, blocking the CD28-CD80/86 co-stimulatory T-cell activation with abatacept, which is a CTLA-4Ig fusion protein, prevented CD4+ T cell activation and reduced the accelerated development of atherosclerosis in hypercholesterolemic ApoE3*Leiden mice^{10,13}. However, these mechanistic studies are still in progress, and future studies are needed to investigate the interplay between the PD-1/PD-L1/CTLA-4 axis and the CRS associated with CAR-T therapy and pyroptosis.

References:

1. Frey, N. & Porter, D. Cytokine Release Syndrome with Chimeric Antigen Receptor T Cell Therapy. *Biol Blood Marrow Transplant* 25, e123-e127, doi:10.1016/j.bbmt.2018.12.756 (2019).
2. Lee, D. W. et al. ASTCT Consensus Grading for Cytokine Release Syndrome and Neurologic Toxicity Associated with Immune Effector Cells. *Biol Blood Marrow Transplant* 25, 625-638, doi:10.1016/j.bbmt.2018.12.758 (2019).
3. Xiao, X. et al. Mechanisms of cytokine release syndrome and neurotoxicity of CAR T-cell therapy and associated prevention and management strategies. *J Exp Clin Cancer Res* 40, 367, doi:10.1186/s13046-021-02148-6 (2021).
4. Liu, Y. et al. Gasdermin E-mediated target cell pyroptosis by CAR T cells triggers cytokine release syndrome. *Sci Immunol* 5, doi:10.1126/sciimmunol.aax7969 (2020).
5. Zhou, Z. et al. Granzyme A from cytotoxic lymphocytes cleaves GSDMB to trigger pyroptosis in target cells. *Science* 368, doi:10.1126/science.aaz7548 (2020).
6. Shi, J. et al. Cleavage of GSDMD by inflammatory caspases determines pyroptotic cell death. *Nature* 526, 660-665, doi:10.1038/nature15514 (2015).
7. Gong, T., Liu, L., Jiang, W. & Zhou, R. DAMP-sensing receptors in sterile inflammation and inflammatory diseases. *Nat Rev Immunol* 20, 95-112, doi:10.1038/s41577-019-0215-7 (2020).
8. Gust, J. et al. Endothelial Activation and Blood-Brain Barrier Disruption in Neurotoxicity after Adoptive Immunotherapy with CD19 CAR-T Cells. *Cancer Discov* 7, 1404-1419, doi:10.1158/2159-8290.Cd-17-0698 (2017).
9. Quagliariello, V. et al. Immune checkpoint inhibitor therapy increases systemic SDF-1, cardiac DAMPs Fibronectin-EDA, S100/Calgranulin, galectine-3, and NLRP3-MyD88-chemokine pathways. *Front Cardiovasc Med* 9, 930797, doi:10.3389/fcvm.2022.930797 (2022).
10. Suero-Abreu, G. A., Zanni, M. V. & Neilan, T. G. Atherosclerosis With Immune Checkpoint Inhibitor Therapy: Evidence, Diagnosis, and Management: JACC: CardioOncology State-of-the-Art Review. *JACC CardioOncol* 4, 598-615, doi:10.1016/j.jaccao.2022.11.011 (2022).
11. Gotsman, I. et al. Proatherogenic immune responses are regulated by the PD-1/PD-L pathway in mice. *J Clin Invest* 117, 2974-2982, doi:10.1172/jci31344 (2007).
12. Bu, D. X. et al. Impairment of the programmed cell death-1 pathway increases atherosclerotic lesion development and inflammation. *Arterioscler Thromb Vasc Biol* 31, 1100-1107, doi:10.1161/atvbaha.111.224709 (2011).
13. Ewing, M. M. et al. T-cell co-stimulation by CD28-CD80/86 and its negative regulator CTLA-4 strongly influence accelerated atherosclerosis development. *Int J Cardiol* 168, 1965-1974, doi:10.1016/j.ijcard.2012.12.085 (2013).

Reviewer #3 (Remarks to the Author):

I have gladly reviewed the manuscript by Karthikeyan et al on the cardiotoxicity of CART-cell products and T-cell engagers in patients with hematological malignancies. The study is based on a WHO-based pharmacovigilance registry (VigiBase), with its many potential biases.

I have a number of comments:

MAJOR:

- The major problem of this study is the bias related to the voluntary decision of a physician to report an adverse or not, and the logical higher probability of reporting severe vs. mild adverse events. In this regard, the "severity" calculations are, by definition biased. One of them particularly caught my attention, which is that hypertriglyceridemia was life-threatening in all patients treated with blinatumomab. This is somehow surprising because hypertriglyceridemia is usually not very relevant as an adverse event and, when it is relevant, it is in the context of HLH-type of complication, which is far more frequent and dangerous in patients receiving CART-cells.

Response: We greatly thank the reviewer for raising a critical point concerning the subjective bias involved with the voluntary reporting of adverse events. We therefore revised the analysis to stick strictly to safety reports from clinical studies to mitigate this bias. As a result of this revision, hypertriglyceridemia no longer appears as a significant signal.

These changes are reflected in the revised manuscript, with the following takeaways from the Results:

Higher Fatality Rates of Hypotension Observed in Tisagenlecleucel Compared to Axicabtagene Ciloleucel in Patients with Diffuse Large B-Cell Lymphoma

In VigiBase, there were 17,479 reports containing the DLBCL indication, which were filtered to 237 clinical study reports for tisagenlecleucel and 42 clinical study reports for axicabtagene ciloleucel for all cardiovascular and investigational adverse reactions. Reports from clinical study were used to mitigate the bias involved from voluntary reporting of severe adverse events.

Although axicabtagene ciloleucel had significantly greater reporting odds ratios (RORs) for hypotension than tisagenlecleucel in DLBCL patients (ROR: 2.54; 95% CI: 1.28-5.03; $p = 0.012$; N_{AD} (axicabtagene ciloleucel) = 18, N_{AD} (tisagenlecleucel) = 54), tisagenlecleucel had a significantly higher fatality rate for hypotension (50.0%) than axicabtagene ciloleucel (5.6%, $p < 0.001$) (Figure 3 and Table 3). Additionally, tisagenlecleucel was associated with cardiac failure ($IC_{025} = 0.366$), with an 85.7% fatality rate. Axicabtagene ciloleucel and tisagenlecleucel had similar reporting for atrial fibrillation in DLBCL patients (ROR: 2.20; 95% CI: 0.56-8.66; $p = 0.22$; N_{AD} (axicabtagene ciloleucel) = 3, N_{AD} (tisagenlecleucel) = 8), with no significant difference in fatality rates [50.0% (tisagenlecleucel) vs 33.3% (axicabtagene ciloleucel)]; $p =$

1.00]. Of the 11 DLBCL CAR-T patients reporting atrial fibrillation, 7 patients (63.6%) were treated with cyclophosphamide/fludarabine combination (Supplementary Table 9).

Tisagenlecleucel is Associated with Fatal Cardiac Arrest and Cardiac Failure in Pediatric Acute Lymphocytic Leukemia

There were 28,803 reports in Vigibase containing the ALL indication, which were filtered to 354 clinical study reports for tisagenlecleucel, 230 clinical study reports for blinatumomab, and 2 clinical study reports for brexucabtagene autoleucel ICSRs for all cardiovascular and investigational adverse reactions.

Tisagenlecleucel is indicated for the treatment of relapsed/refractory ALL in pediatric patients. Blinatumomab, while commonly used for treating relapsed/refractory ALL in adult patients, is also indicated for treatment of pediatric relapsed/refractory ALL. Consequently, safety reports from clinical studies were compared between tisagenlecleucel and blinatumomab for patients aged 17 or younger, resulting in 219 safety reports for tisagenlecleucel and 28 safety reports for blinatumomab for all cardiovascular and investigational adverse reactions.

While blinatumomab had significantly higher reporting for pericardial effusion compared to tisagenlecleucel (ROR: 8.64; 95% CI: 1.65-45.12; $p = 0.021$; N_{AD} (tisagenlecleucel) = 3, N_{AD} (blinatumomab) = 3), the fatality rates were not significantly different [100% (tisagenlecleucel) vs 66.7% (blinatumomab); $p = 1.00$] (Figure 4 and Table 5). Additionally, tisagenlecleucel-associated cardiac arrest and cardiac failure had high fatality rates of 100% and 80.0% respectively. Blinatumomab and tisagenlecleucel had similar reporting for hypotension (ROR: 0.44; 95% CI: 0.16-1.22; $p = 0.13$; N_{AD} (tisagenlecleucel) = 72, N_{AD} (blinatumomab) = 5) as well as similar fatality rates [34.7% (tisagenlecleucel) vs 20.0% (blinatumomab); $p = 0.66$] in pediatric ALL patients. Similarly, tisagenlecleucel and blinatumomab had comparable fatality rates for tachycardia [47.1% (tisagenlecleucel) vs 50.0% (blinatumomab); $p = 1.00$]. Of the 72 pediatric ALL patients treated with tisagenlecleucel who reported having hypotension, about 83.3% were treated with cyclophosphamide/fludarabine combination (Supplementary Table 10).

- I am also shocked to see that not all patients receiving CART-cells also received FluCy chemotherapy. Most if not all clinical trials and protocols involving CART-cells generally include FluCy lymphodepleting chemotherapy, which is not the case for bispecific.

Response: We thank the reviewer for bringing up this important point. By sticking to safety reports from clinical studies, the percentage of patients on fludarabine/cyclophosphamide lymphodepleting chemotherapy significantly rose to 63.6% for the 11 DLBCL patients reporting atrial fibrillation and 83.3% for the 72 pediatric ALL patients reporting hypotension. These results concur with the reviewer's point that most clinical trials involving CAR-T cells generally include fludarabine/cyclophosphamide lymphodepleting chemotherapy. These changes are reflected in the revised manuscript, as shown from the Discussion below:

In both DLBCL CAR-T patients reporting atrial fibrillation and pediatric ALL CAR-T patients reporting hypotension, most of the patient populations (63.6% for DLBCL and 83.3% of ALL) were treated with the chemotherapy regimen of cyclophosphamide/fludarabine.

- Comparing blina vs. tisa-cel is not really fair because the approved indication for tisa-cel is pediatric ALL and blina is mostly used in adult patients.

Response: We thank the reviewer for raising an important point concerning the patient populations involved in blinatumomab and tisagenlecleucel. While blinatumomab is commonly used in adult patients, it is also indicated for the treatment of pediatric ALL. We therefore restricted the comparison of blinatumomab and tisagenlecleucel in pediatric patients aged 17 or younger, as VigiBase includes demographic reports for the 18-44 age group. These changes are reflected in the Results shown below:

Tisagenlecleucel is indicated for the treatment of relapsed/refractory ALL in pediatric patients ¹. Blinatumomab, while commonly used for treating relapsed/refractory ALL in adult patients, is also indicated for treatment of pediatric relapsed/refractory ALL ^{2,3}. Consequently, safety reports from clinical studies were compared between tisagenlecleucel and blinatumomab for patients aged 17 or younger, resulting in 219 safety reports for tisagenlecleucel and 28 safety reports for blinatumomab for all cardiovascular and investigational adverse reactions.

References:

1. Maude, S. L. et al. Tisagenlecleucel in Children and Young Adults with B-Cell Lymphoblastic Leukemia. *New England Journal of Medicine* 378, 439-448, doi:10.1056/NEJMoa1709866 (2018).
2. von Stackelberg, A. et al. Phase I/Phase II Study of Blinatumomab in Pediatric Patients With Relapsed/Refractory Acute Lymphoblastic Leukemia. *J Clin Oncol* 34, 4381-4389, doi:10.1200/jco.2016.67.3301 (2016).
3. Locatelli, F. et al. Effect of Blinatumomab vs Chemotherapy on Event-Free Survival Among Children With High-risk First-Relapse B-Cell Acute Lymphoblastic Leukemia: A Randomized Clinical Trial. *Jama* 325, 843-854, doi:10.1001/jama.2021.0987 (2021).

MINOR:

- ICSR is not spelled out until much later, probably because the order of the paper was different in a previous submission (Methods before Results).

Response: We apologize for the oversight and corrected this in the Results.

Based on the records obtained up to January 26, 2022, a total of 30,039,123 Individual Case Safety Reports (ICSRs) were reported in VigiBase.

- The manuscript is written in such a way that all the knowledge (all the "truths") seem to come from the tables and the figures. "From table X we conclude that..." "Table X showed that tisa-cel had a significantly higher fatality rate". I would somehow recommend to rephrase most of these sentences.

Response: The sentences were rephrased to account for the reviewer's feedback.

REVIEWERS' COMMENTS:

Reviewer #1 (Remarks to the Author):

The manuscript entitled “Cardiotoxic Profiles of CAR-T Therapy and Bispecific T-Cell Engagers in Hematological Cancers” is very interesting in the field of cardioncology.

This is a good and very clear article that allows the readers to understand quickly the topic. The figures and the tables are of a good quality, furthermore, methods and results are coherent with the initial hypothesis of the authors.

After reading the article, that was modified according to the changes required by reviewers' proposals, in my opinion it is suitable for publication in this journal.

Reviewer #2 (Remarks to the Author):

Manuscript titled "Cardiotoxic Profiles of CAR-T Therapy and Bispecific T-Cell Engagers in Hematological Cancers" is a very interesting article in the field of cardioncology. The overall structure is of good quality and easy to read. Methods and results are clear and results corroborate the initial hypothesis of the authors. Figures and tables are of sufficient quality and easy to read as well as to understand to readers. I suggest some improvements, specifically in introduction and/or discussion.

In introduction, authors should describe the key role of Myd-88 and NLRP-3 pathways in anticancer functions of immunotherapy, including immune checkpoint inhibitors and CAR-T and their role also in vascular and myocardial affection, such as myocarditis and vasculitis (cite doi: 10.3389/fcvm.2022.930797). ICIs therapy increases the risk of atherosclerosis in cancer patients through the involvement of immune-based inflammation in vascular tissue. Authors should highlight this point and discuss the differences with CAR-T therapy. Based on these changes, the article could be suitable for publication in this journal.

Reviewer #3 (Remarks to the Author):

The authors have adequately addressed all my comments. Many thanks and congratulations.